# IL-18/IL-37/IP-10 signalling complex as a potential biomarker for discriminating active and latent TB

Sebastian Wawrocki[1], Michal Seweryn[2], Grzegorz Kielnierowski[3], Wieslawa Rudnicka[1], Marcin Wlodarczyk[1], Magdalena Druszczynska[1]*

1 Department of Immunology and Infectious Biology, Institute of Microbiology, Biotechnology and Immunology, Faculty of Biology and Environmental Protection, University of Lodz, Banacha 12/16, Poland,
2 Center for Medical Genomics OMICRON, Jagiellonian University, Medical College, Cracow, Poland,
3 Regional Specialized Hospital of Tuberculosis, Lung Diseases and Rehabilitation, Szpitalna 5, Tuszyn, Poland

* magdalena.druszczynska@biol.uni.lodz.pl

## Abstract

### Background

Currently, there are serious limitations in the direct diagnosis of active tuberculosis (ATB). We evaluated the levels of the IL-18/IL-37/IP-10 signalling complex proteins in *Mycobacterium tuberculosis* (*M.tb*)-specific antigen-stimulated QuantiFERON® Gold In-Tube (QFT) cultures and in serum samples from ATB patients, healthy individuals with latent *M.tb* infection (LTBI) and healthy controls (HC) to examine whether combined analyses of these proteins were useful in the differentiation of *M.tb* states.

### Methods

The concentrations of IL-18, IL-18BP, IFN-γ, IL-37 and IP-10 in the serum and QFT supernatants were measured using specific enzyme-linked immunosorbent assay (ELISA) kits. Free IL-18 levels were calculated using the law of mass action.

### Results

Increased concentrations of total and free IL-18, IL-18BP, IFN-γ and IP-10 in the sera of ATB patients were detected. These increases were not counterbalanced by the overproduction of IL-37. Complex co-expression of serum IL-18BP and IL-37, IP-10 and IFN-γ was identified as the highest discriminative biomarker set for the diagnosis of ATB.

### Conclusions

Our results suggest that the IL-18 signalling complex might be exploited by *M. tuberculosis* to expand the clinical manifestations of pulmonary TB. Therefore, direct analysis of the serum components of the IL-18/IL-37 signalling complex and IP-10 may be applicable in designing novel diagnostic tests for ATB.

**Data Availability Statement:** All relevant data are within the article and its Supporting Information files.

**Funding:** This work was supported by the National Science Centre grants no 2015/19/N/NZ6/01385 and 2016/21/B/NZ7/01771.

**Competing interests:** The authors have declared that no competing interests exist.

## Introduction

Tuberculosis (TB) affects approximately 10 million people causing 2 million deaths annually [1]. Approximately 1/3 of the human population is infected with *Mycobacterium tuberculosis* (*M.tb*), the causative agent of TB, and 5–10% of this population develop active tuberculosis (ATB) disease during their lifetime. The remaining 90–95% of individuals mount an immune response and develop latent tuberculosis infection (LTBI) [2]. TB is predominantly a disease of the lungs and the transmission of *M.tb* bacilli occurs through airborne granulomatous particles released into the air by individuals suffering from pulmonary ATB. Tuberculous granulomas are formed in infected lungs and are aggregates of macrophages surrounded by a lymphocyte marginal zone that encloses the infecting mycobacteria [3]. *M.tb* persists in a dormant state inside macrophages for long periods of time. The immune status of the host's macrophages and T cells and the activation of cytokines, primarily IFN-γ, control the granuloma structure and *M.tb* replication in asymptomatic LTBI. Due to an unpredictable reason, the bacilli reactivate in 5–10% of LTBI subjects, and caseous granulomas develop and initiate a clinical disease and the spread of virulent bacilli in the environment. The risk of progression from LTBI to active TB may be increased by some factors such as HIV infection, chronic renal failure, diabetes, organ transplantation or therapy using tumor necrosis factor-alpha blockers. The identification of *M.tb*-infected individuals and the appropriate treating those who develop ATB and preventing those with an increased risk of TB progression are undoubtedly crucial for effective TB control. Currently, there are limitations in the direct ATB detection via the microscopic visualization of acid-fast bacteria in the sputum and observation of *M.tb* growth in long-term cultures. Interferon-gamma release assays (IGRAs) are used to diagnose LTBI. These tests measure the release of IFN-γ in response to *M.tb*-specific antigens in whole blood cultures. However, IGRAs in combination with tuberculin skin tests are not sufficiently accurate to diagnose ATB [1,4]. To satisfy the demand for rapid and accurate TB diagnostic tests, we performed analyses of several classification algorithms to rank proteins of the IL-18/IL-37/IP-10 signalling complex according to their usefulness in the differentiation of *M.tb* infection states. Progression towards tuberculosis disease correlates with the loss of organization in the granulomas [5]. The immunological and inflammatory environments of granulomas change due to the recirculation of immune cells and the release of cytokines that reach the periphery. Therefore, evaluations of cytokine/chemokine profiles in the blood are promising for the differentiation of infectious states in TB [6–8].

The cytokine IL-18 is implicated in the protective and pathological processes of *M.tb* infection [9–11]. The activity of IL-18 occurs via an IL-18 binding receptor (IL-18R) complex formed by two chains: a ligand-binding alpha chain (IL-18Rα), and a beta chain (IL-18Rβ), which is responsible for the induction of a proinflammatory signal [12]. The formation of an IL-18Rα/IL-18Rβ heterodimer triggers the signalling cascade that leads to activation of the transcription factor NF-κβ [13]. The excessive inflammatory signalling of IL-18 is reduced by a constitutively secreted IL-18 binding protein (IL-18BP), which neutralizes circulating IL-18 to lower free IL-18 compared to total IL-18 levels. Therefore, the production of IFN-γ and other proinflammatory cytokines is reduced [14]. IL-18BP also shows a high affinity for IL-37, which is an inhibitor of the innate inflammatory responses involved in curbing excessive inflammation [15,16]. After binding to IL-18BP, IL-37 subsequently binds IL-18Rβ, which inhibits the proinflammatory activity of IL-18 [17,18]. We performed a combined analysis of the proteins, free and total IL-18, IL-18BP, IL-37, IFN-γ in QFT supernatants and directly in serum samples from pulmonary TB patients and healthy individuals with or without latent *M.tb* infection to identify new markers for the diagnosis of ATB. In parallel with the proteins' analysis, we measured serum IFN-γ-inducible protein 10 (IP-10/CXCL10) levels, a chemokine mediating leukocyte recruitment and activation.

## Materials and methods

### Study population

A total population of 238 *M. bovis Bacillus Calmette-Guerin* (BCG)-vaccinated adults of both genders, 18–81 years of age, was enrolled in the present study. All participants were unrelated Poles who signed a formal written consent for the use of their blood for research purposes. The study protocol complied with the most recent Declaration of Helsinki, and the Ethics Committee of the University in Lodz, Poland approved the protocol. The study cohort included 95 patients with active pulmonary TB (ATB), which was microbiologically confirmed (51) or not confirmed (44) in a triple sputum culture. They were recruited from the Regional Specialised Hospital of Tuberculosis, Lung Diseases and Rehabilitation in Tuszyn, Poland. A full history was taken from all patients, and experienced physicians completed general and clinical examinations. The final diagnosis was based on the clinical symptoms, a chest X-ray image, microscopic and microbiological evaluations of sputum samples and a proper response to anti-tuberculous treatment. Blood samples were taken prior to the start of therapy. Healthy individuals (143) entering the study were classified as *M.tb*-infected (LTBI) (52) or *M.tb*-uninfected (91) based on the interferon-gamma release assay results (QuantiFERON-TB Gold Plus; QFT). None of the healthy volunteers had a history of TB.

The age, sex, BCG vaccination status as well as tuberculin skin test (TST) and QFT results of the study participants are summarized in Table 1. The group of the study comprised 95 patients diagnosed with active pulmonary TB (ATB), microbiologically confirmed (51) or not confirmed (44) by a triple sputum culture, 52 individuals without TB history, who were latently infected with *M.tb* (LTBI), and 91 healthy controls without *M.tb* infection (HC). There were no significant differences between studied groups regarding age or BCG vaccination rate. The proportion of men in the ATB group (56%) was significantly higher than in LTBI (31%) and HC (31%) groups (p<0.05). Fifty one (54%) ATB patients had a positive sputum culture for *M.tb*. Forty four (46%) *M.tb* culture negative patients were diagnosed on the basis of clinical manifestations, typical features' on chest radiographs and proper response to anti-tuberculous treatment. Fifty-five % of ATB patients exhibited a positive TST result with an induration diameter of more than 10 mm, whereas a positive QFT result was found in 61% of individuals from this group. Both subgroups of ATB patients were characterized by a similar proportion of TST-positive and QFT-positive results. Nine out of 95 (9%) ATB patients had a history of healed pulmonary TB. Six percent of ATB patients suffered from diabetes or chronic renal failure, whereas cardiovascular or neurological diseases were diagnosed in 15% and 3% patients, respectively.

### Blood samples

A 5-ml volume of venous blood samples was used to prepare the sera and perform a QuantiFERON-TB® Gold Plus test (QFT, Qiagen, Hilden, Germany). Blood was collected in four 1-ml tubes (Nil control, TB antigen-specific 1 (TB1), TB antigen-specific 2 (TB2), Mitogen control), and IFN-γ levels in the supernatants were measured immunoenzymatically after a 24-hour incubation.

### IL-18, IL-18BP, IL-37, IFN-γ, IP-10 estimation and calculation of free IL-18

The concentrations of IL-18, IL-18BP and IL-37 in the sera and QFT cell-free culture supernatants were determined using commercially available specific ELISA kits: Human Total IL-18 DuoSet ELISA (R&D, Minneapolis, USA), Human IL-18BPa DuoSet ELISA (R&D) and Human IL-37/IL-1F7 Duoset ELISA (R&D). IFN-γ and IP-10 concentrations in sera were

**Table 1. Characteristics of the study participants.**

|  | ATB | LTBI | HC |
|---|---|---|---|
| N | 95 | 52 | 91 |
| Ethnicity | Caucasians | Caucasians | Caucasians |
| Country of origin | Poland | Poland | Poland |
| Sex, N (%) |  |  |  |
| Men | 53 (56%) | 16 (31%)[a] | 28 (31%)[b] |
| Women | 42 (44%) | 36 (69%) | 63 (69%) |
| Age |  |  |  |
| Median | 51 | 52 | 41 |
| range | 19–81 | 20–81 | 18–75 |
| years (IQR) | 33–63 | 41–56 | 33–52 |
| BCG vaccination, N (%) | 95 (100%) | 52 (100%) | 91 (100%) |
| TST induration diameter (mm) |  |  |  |
| median | 12 |  |  |
| range | 0–35 |  |  |
| TST result, N (%) |  | n.d | n.d |
| positive | 52 (55%) |  |  |
| negative | 43 (45%) |  |  |
| QFT result, N (%) |  |  |  |
| positive | 58 (61%) | 52 (100%) | 0 (0%) |
| negative | 37 (39%) | 0 (0%) | 91 (100%) |
| *M.tb* culture |  |  |  |
| positive | 51 (54%) | n.d | n.d |
| negative | 44 (46%) |  |  |
| History of healed ATB | 9 (9%) | 0 (100%) | 0 (100%) |

Abbreviations: ATB–active tuberculosis, HC- healthy controls, LTBI–latent *M.tb* infection, BCG–Bacillus Calmette-Guerin, QFT–QuantiFERON TB Gold Plus, TST-tuberculin skin test, IQR—interquartile range, n.d–not determined.

There was a significant difference among studied groups

[a]male LTBI vs ATB (p<0.05) and

[b]male HC vs ATB (p<0.05) ($\chi^2$ test).

assessed using Human IFN-γ Duoset ELISA (R&D) and Human IP-10 Duoset ELISA (R&D). The law of mass action was used to calculate the level of free IL-18 [19]. It is known that one IL-18BP molecule binds a single molecule of IL-18, and this interaction has a dissociation constant (Kd) of 0.4 nM. Therefore, the level of free IL-18 was calculated from the equation $x = (-b \pm (b^2-4ac)^{1/2})/2a$, where x is [IL-18free], b is = [IL-18BP]–[IL-18] + Kd, and c is -Kd·IL-18[13,19,20]. The levels of the studied proteins in the QFT supernatants were calculated after subtraction of baseline levels obtained from NIL tube.

## Statistical analyses

Statistical analyses were performed using Statistica 12 PL (Statsoft, Poland). Comparisons of the frequencies were tested using the $\chi^2$ or Fisher's exact test. Differences in the levels of the studied proteins were analysed using the non-parametric Kruskal-Wallis test. Differences with $p < 0.05$ were considered statistically significant.

Target statistical and machine-learning methods were used as implemented in R. 'pROC', 'ROCR' packages as well as custom codes (available upon request) were used in the ROC analysis. For the classical analysis of association between levels of expression of single proteins and

protein ratios logistic regression was used in order to make the comparison of performance of single markers to all available markers (full logistic model) 'fair'. The AUC values were estimated via 5-fold cross-validation and based on at least 500 bootstrap replicates. The Random Forest algorithm was used as implemented in the 'randomForest' package. Pearson's correlation coefficient was used for co-expression analyses. For the network analysis, the custom R code was used (available upon request), and data discretization for the estimation of Renyi divergence was performed using the 'infotheo' and 'equalfreq' options. The t-SNE algorithm was used as implemented in 'Rtsne'. The ordinal elastic-net algorithm was applied as implemented in 'ordinalNet'.

## Results

### Serum IL-18, IL-18BP, IL-37, IFN-γ and IP-10 levels in the studied groups

The concentration of total IL-18 in the sera from the ATB patients (median (Me) 568.1 (IQR 371.9, 980.3) pg/ml) was significantly higher than those found in the LTBI (Me 261.6 (IQR 109.3, 492.3) pg/ml) and HC (Me 283.1 (IQR 177.5, 441.4) pg/ml) groups (p<0.001) (Fig 1A).

Total IL-18 levels were increased in both culture-positive (Me 719.5 (IQR 416.3, 1146.0) pg/ml) and culture-negative (Me 444.2 (IQR 305.8, 828.1) pg/ml) ATB patients (data not shown). Similarly to the total IL-18 level, the serum concentration of IL-18BP was significantly higher in ATB patients (Me 43.5 (IQR 31.7, 60.4) ng/ml, culture-positive (Me 44.9 (IQR 32.5, 59.3) ng/ml, culture-negative (Me 43.4 (IQR 27.8, 77.2) ng/ml) than in LTBI (Me 28.8 (IQR 21.1, 42.8) ng/ml; p<0.001) or HC (Me 32.7 (IQR 22.7, 43.7) ng/ml; p<0.001) groups (Fig 1B).

The serum concentration of free IL-18 calculated from IL-18 and IL-18BP levels was significantly higher in ATB patients (Me 5.5 (IQR 3.0, 10.5) pg/ml; culture-positive (Me 5.5 (IQR 3.7, 10.9) pg/ml, culture-negative (Me 3.6 (IQR 2.1, 8.3) pg/ml) than LTBI (Me 3.6 (IQR 1.7, 5.5)

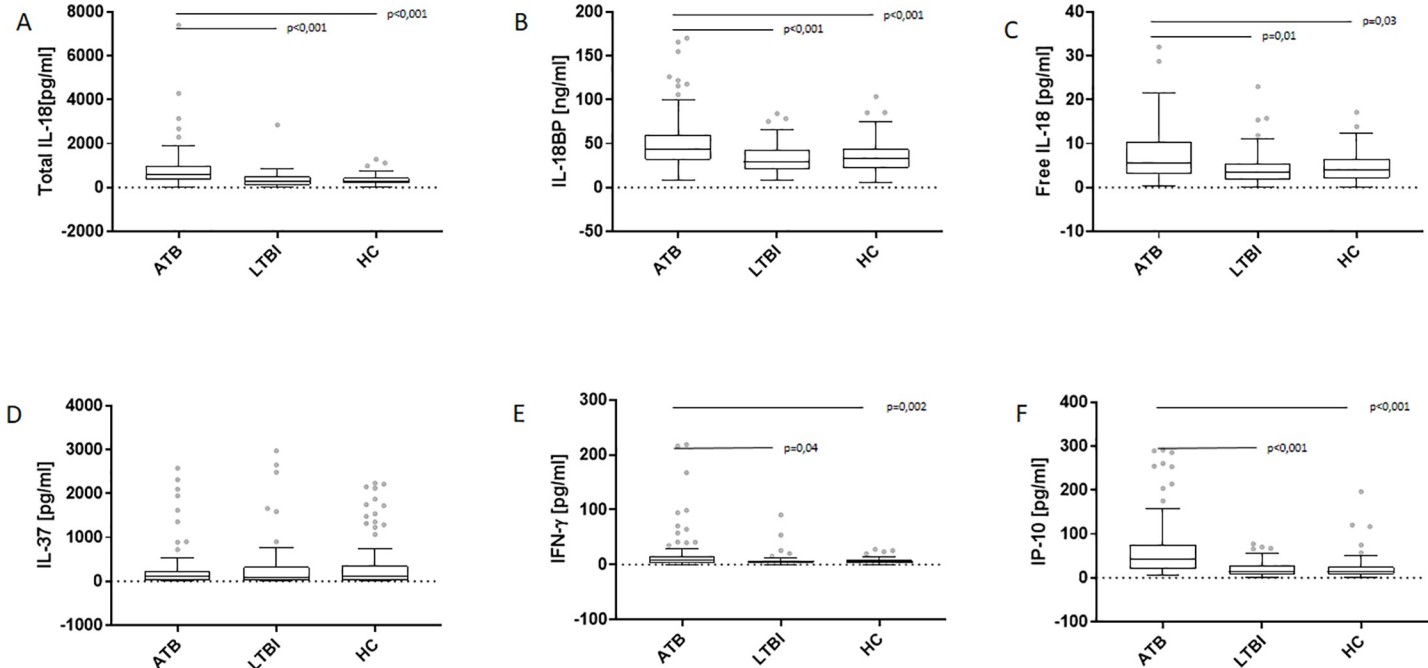

**Fig 1. Serum levels of total and free IL-18, IL-18BP, IL-37, IFN-γ and IP-10 in the groups of the study.** Boxplots with median (horizontal line within the box), interquartile range (box limits), and extremes (whiskers) of serum levels of total IL-18 (A), IL-18BP (B), free IL-18 (C), IL-37 (D), IFN-γ (E) and IP-10 (F) in the groups of patients with active tuberculosis (ATB) and latent *M.tb* infection (LTBI) and healthy controls (HC).

pg/ml; p = 0.012) or HC (Me 4.0 (IQR 2.1, 6.5) pg/ml; p = 0.036) groups (Fig 1C). In contrast, the serum concentrations of IL-37 were similar among studied groups: ATB (Me 102.6 (IQR 12.6, 236.6) pg/ml, LTBI (Me 78.4 (IQR 23.3, 323.3) pg/ml, HC (Me 103.5 (IQR 9.8, 357.3) pg/ml (Fig 1D). On the other hand, the level of IFN-γ, in the ATB patients' sera was significantly higher (Me 6.7 (IQR 3.5, 15.0) pg/ml; culture-positive (Me 8.8 (IQR 4.8, 20.4) pg/ml, culture-negative (Me 5.7 (IQR 3.2, 12.2) pg/ml) than that observed in the LTBI (Me 5.0 (IQR 3.3, 7.5) pg/ml; p = 0.049) or HC (Me 5.7 (IQR 3.0, 8.4) pg/ml; p = 0.002) individuals (Fig 1E). As shown in Fig 1F, the serum level of IP-10 was significantly higher in the ATB group (Me 43.5 (IQR 19.9, 75.7) pg/ml; culture-positive (Me 49.3 (IQR 27.4, 83.9) pg/ml, culture-negative (Me 35.8 (IQR 13.6, 57.2) pg/ml) than in the groups of LTBI (Me 13.7 (IQR 8.2, 28.0) pg/ml) and HC (Me 14.0 (IQR 7.8, 25.3) pg/ml) (p<0.001). There were no significant differences in the serum levels of the total IL-18, IL-18BP, free IL-18, IL-37, IFN-γ and IP-10 between ATB patients with positive or negative QFT and TST results (Fig 2A–2F).

Analysis of the area under the ROC revealed that the full logistic model resulted in an AUC = 0.82 (CI = (0.82; 0.83)) for the HC vs. ATB comparison, an AUC = 0.65 (CI = (0.61; 0.70)) for HC vs. LTBI, and an AUC = 0.79 (CI = (0.73; 0.80)) for LTBI vs. ATB. Table 2 shows the AUCs for individual proteins.

The analysis of dependence between the levels of all proteins revealed certain correlations, that were specific for two groups of the study (IL-18BP and IP-10 as well as IL-18 and IL-18BP for ATB and LTBI) and some relationships specific only for certain study groups (e.g. IL-18 and IL-37 or il-18 and IP-10 for HC, IL-18 and IFN-γ or IFN-γ and IP-10 for ATB, and no correlation specific for LTBI) (Table 3).

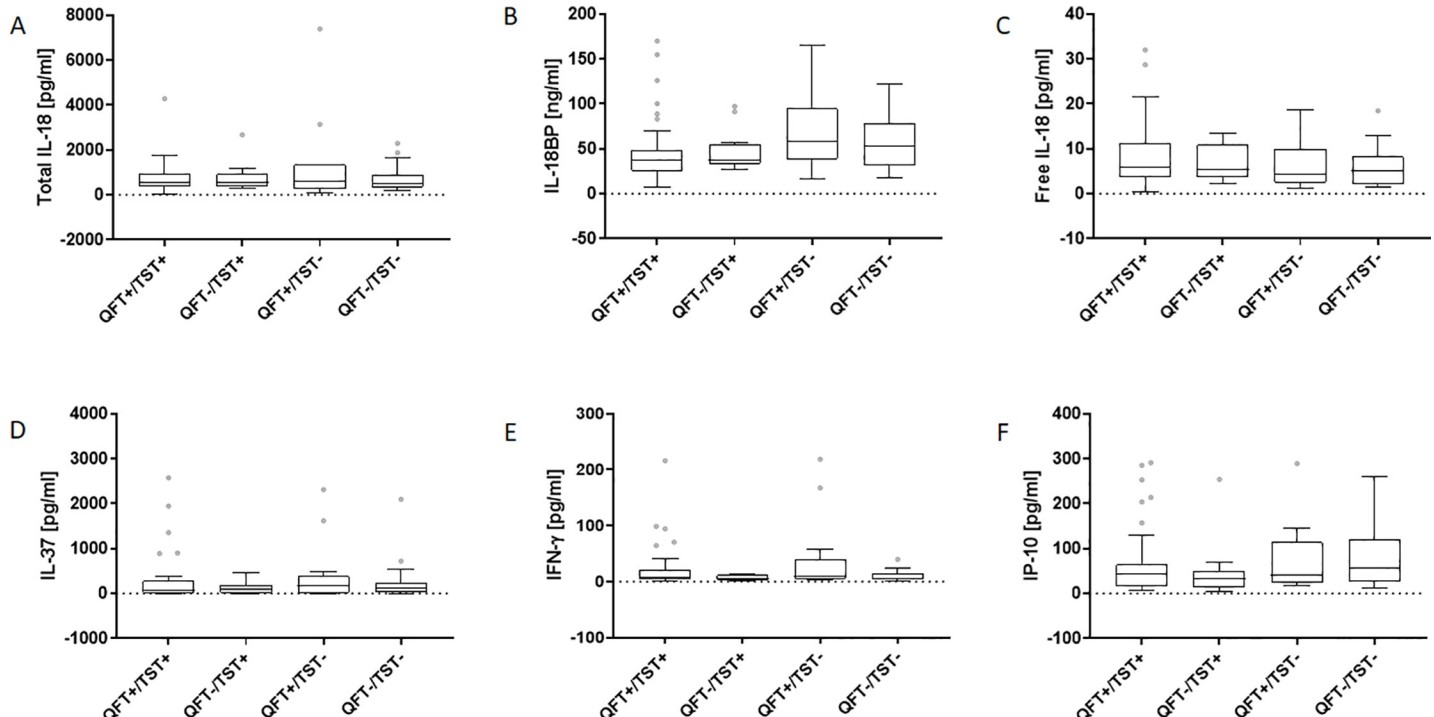

**Fig 2. Serum levels of total IL-18, IL-18BP, free IL-18, IL-37, IP-10 and IFN-γ versus QFT/TST results.** Boxplots with median (horizontal line within the box), interquartile range (box limits), and extremes (whiskers) of serum levels of total IL-18 (A), IL-18BP (B), free IL-18 (C), IL-37 (D), IFN-γ (E) and IP-10 (F) in the group of active tuberculosis patients with positive or negative QuantiFERON (QFT) and tuberculin skin test (TST) results.

**Table 2. Predictive values (median AUC and 95% CI) of individual protein levels measured in serum.**

| Parameter | HC vs. ATB | HC vs. LTBI | LTBI vs. ATB |
|---|---|---|---|
| | median AUC (95% CI) | | |
| IL-18 | 0.76 (0.76;0.77) | 0.71 (0.65;0.77) | 0.76 (0.75;0.77) |
| IL-18BP | 0.67 (0.66;0.67) | 0.62 (0.56;0.69) | 0.69 (0.68;0.70) |
| IL-37 | 0.59 (0.52;0.59) | 0.76 (0.69;0.82) | 0.64 (0.59;0.69) |
| IFN-γ | 0.57 (0.56;0.58) | 0.61 (0.58;0.67) | 0.52 (0.50;0.56) |
| IP-10 | 0.78 (0.77;0.78) | 0.81 (0.75;0.86) | 0.75 (0.75;0.76) |

ATB–active tuberculosis (n = 95), HC- healthy controls (n = 91), LTBI–latent *M.tb* infection (n = 52)

### *M.tb* antigens–stimulated IL-18, IL-18BP, IL-37 and IFN-γ levels in QFT supernatants

As shown in Fig 3A, the *M.tb* antigens-stimulated levels of total IL-18 were significantly increased in the patients with ATB (Me 765.5 (IQR 431.6, 1122.0) pg/ml, culture-positive (Me 810.0 (IQR 440.4, 1142.0) pg/ml, culture-negative (Me 734.0 (IQR 403.7, 1131.0) pg/ml) compared to those of the LTBI (Me 445.7 (IQR 29.6, 656.1) pg/ml; p<0.001) or the HC group (Me 397.5 (IQR 222.8, 511.6) pg/ml; p<0.001). The level of IL-18BP was significantly higher (p<0.001) in ATB patients (Me 43.1 (IQR 32.5, 63.8) ng/ml) than HC (Me 32.2 (IQR 18.5, 50.9) ng/ml) (Fig 3B). There were no significant differences in the levels of free IL-18 or IL-37 in the QFT supernatants among studied groups (Fig 3C and 3D). The IFN-γ concentration in QFT supernatants was significantly higher in LTBI (Me 91.8 (IQR 37.5, 262.0) pg/ml) than ATB (Me 39.3 (IQR 13.0, 144.2) pg/ml, culture-positive (Me 49.6 (IQR 18.6, 144.2) pg/ml, culture negative (Me 25.3 (IQR 10.8, 159.0) pg/ml; p = 0.003) or HC groups (Me 7.9 (IQR 6.3, 10.6) pg/ml; p<0.001) (Fig 3E). However, there was no association between the results of QFT or TST and the levels of IL-18, IL-18BP or IL-37 measured in the QFT supernatants (Fig 4A–4D).

Analysis based on the area under the ROC showed full logistic model of an AUC = 0.91 (CI = (0.90; 0.91)) for the HC vs. ATB comparison, an AUC = 0.98 (CI = (0.95; 0.99)) for HC vs. LTBI, and an AUC = 0.70 (CI = (0.69; 0.71)) for LTBI vs. ATB. Table 4 shows the discriminative power for the levels of individual proteins.

**Table 3. Correlation (r) and p values between protein levels in serum.**

| Parameters | HC | | LTBI | | ATB | |
|---|---|---|---|---|---|---|
| | n = 91 | | n = 52 | | n = 95 | |
| | r | p | r | p | r | p |
| IL-18 ~ IL-18BP | 0.021 | 0.765 | 0.258 | 0.006 | 0.180 | 0.009 |
| IL-18 ~ IL-37 | -0.130 | 0.070 | -0.012 | 0.899 | -0.005 | 0.940 |
| IL-18 ~ IFN-γ | 0.025 | 0.718 | -0.100 | 0.293 | 0.210 | 0.002 |
| IL-18 ~ IP-10 | 0.152 | 0.033 | 0.154 | 0.107 | 0.112 | 0.107 |
| IL-18BP ~ IL-37 | 0.093 | 0.195 | -0.122 | 0.203 | -0.106 | 0.128 |
| IL-18BP ~ IFN-γ | 0.048 | 0.495 | 0.043 | 0.647 | 0.186 | 0.007 |
| IL-18BP ~ IP-10 | 0.100 | 0.163 | 0.188 | 0.049 | 0.239 | <0.001 |
| IL-37 ~ IFN-γ | 0.081 | 0.255 | 0.046 | 0.629 | 0.003 | 0.956 |
| IL-37 ~ IP-10 | 0.018 | 0.796 | -0.089 | 0.350 | 0.042 | 0.546 |
| IFN-γ ~ IP-10 | 0.066 | 0.355 | 0.086 | 0.364 | 0.263 | <0.001 |

ATB–active tuberculosis, HC- healthy controls, LTBI–latent *M.tb* infection

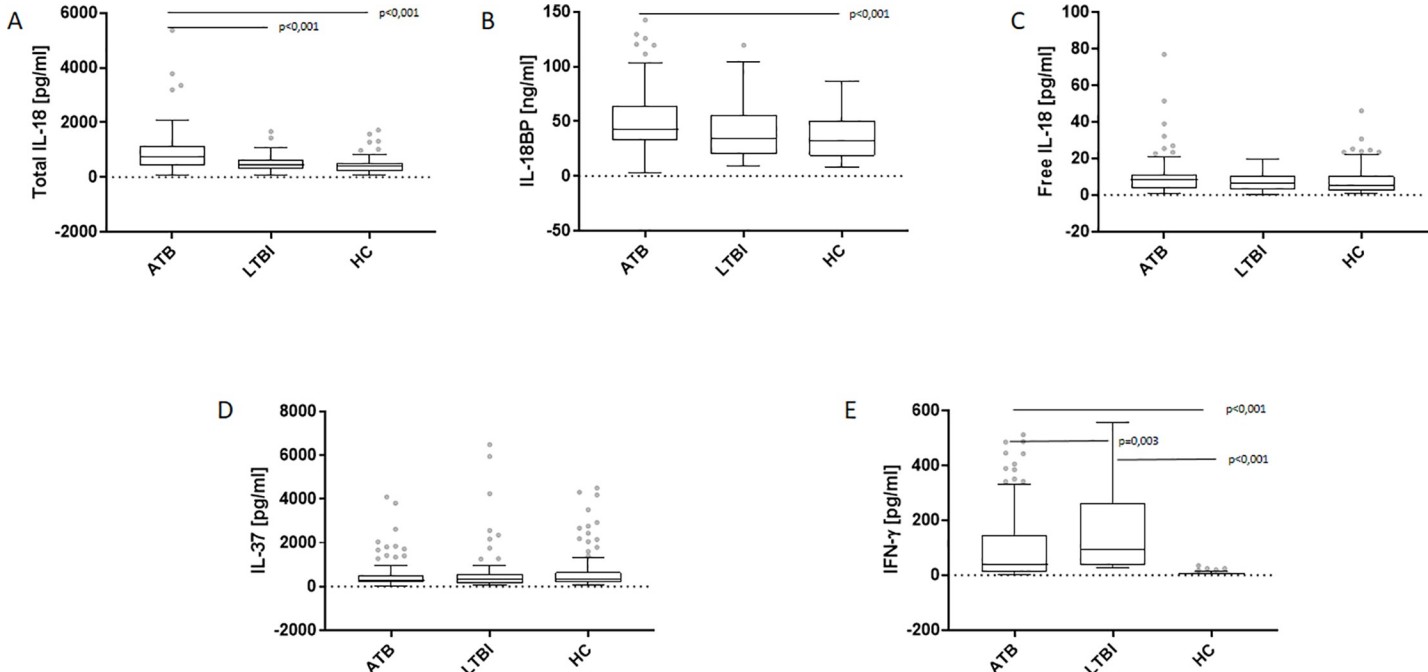

**Fig 3. Total IL-18, IL-18BP, free IL-18, IL-37 and IFN-γ levels in *M.tb* antigens-stimulated QuantiFERON (QFT) supernatants.** Boxplots with median (horizontal line within the box), interquartile range (box limits), and extremes (whiskers) of levels of total IL-18 (A), IL-18BP (B), free IL-18 (C), IL-37 (D) and IFN-γ (E) in *M.tb* antigens-stimulated QFT supernatants in the groups of patients with active tuberculosis (ATB) and latent *M.tb* infection (LTBI) and healthy controls (HC).

Correlation analyses revealed pairs of proteins that were specific for certain study groups (IL-18 and IFN-γ and IL-18 and IL-37 for HC; IL-18BP and IL-37 for LTBI; and IL-18 and IL-18BP for ATB). A summary of these results is provided in Table 5.

## Protein ratios in the sera and QFT supernatants

The discriminative powers of the ratios of serum and QFT supernatants between any two proteins of the IL-18 signalling complex were analysed. The highest discriminative powers in the sera (1) were IL-18/IL-37 for HC and ATB (AUC = 0.69, CI = (0.68; 0.69)), (2) IL-18/IL-18BP for HC and LTBI (AUC = 0.82, CI = (0.76; 0.88)) and (3) IL-18/IL-18BP for LTBI and ATB (AUC = 0.66, CI = (0.64; 0.67)). On the contrary, the highest discriminative powers in the QFT supernatants were (1) IL-37/IFN-γ for HC and ATB (AUC = 0.75, CI = (0.74; 0.76)), (2) IL-18BP/IFN-γ for HC and LTBI (AUC = 0.91, CI = (0.90; 0.91)), and (3) IL-18/IFN-γ for LTBI and ATB (AUC = 0.72, CI = (0.71; 0.74)).

## Selection of the most informative protein ratios using random forest

An approach based on the random forest, which is a sample classification and feature selection algorithm, returned an importance score for each ratio of protein levels and an optimal number of features to be used in classifications estimated using cross-validation. Even with no statistically significant differences between studied groups, the IL-18, IL-18BP and IFN-γ from QFT supernatants were most informative (in terms of AUC) in the LTBI vs. HC comparison. The IL-18/IL-37, IL-18/IFN-γ, IL-18BP/IFN-γ and IL-37/IFN-γ protein ratios had the highest importance scores, and the levels of IL-18, IL-18BP and IFN-γ and the ratios between IL-18BP/IFN-γ and IL-37/IFN-γ remained informative in joint analyses. IL-18, IL-18BP and IFN-γ were most informative for the ATB vs. HC comparison, and the IL-18/IL-37, IL-18/IFN-γ,

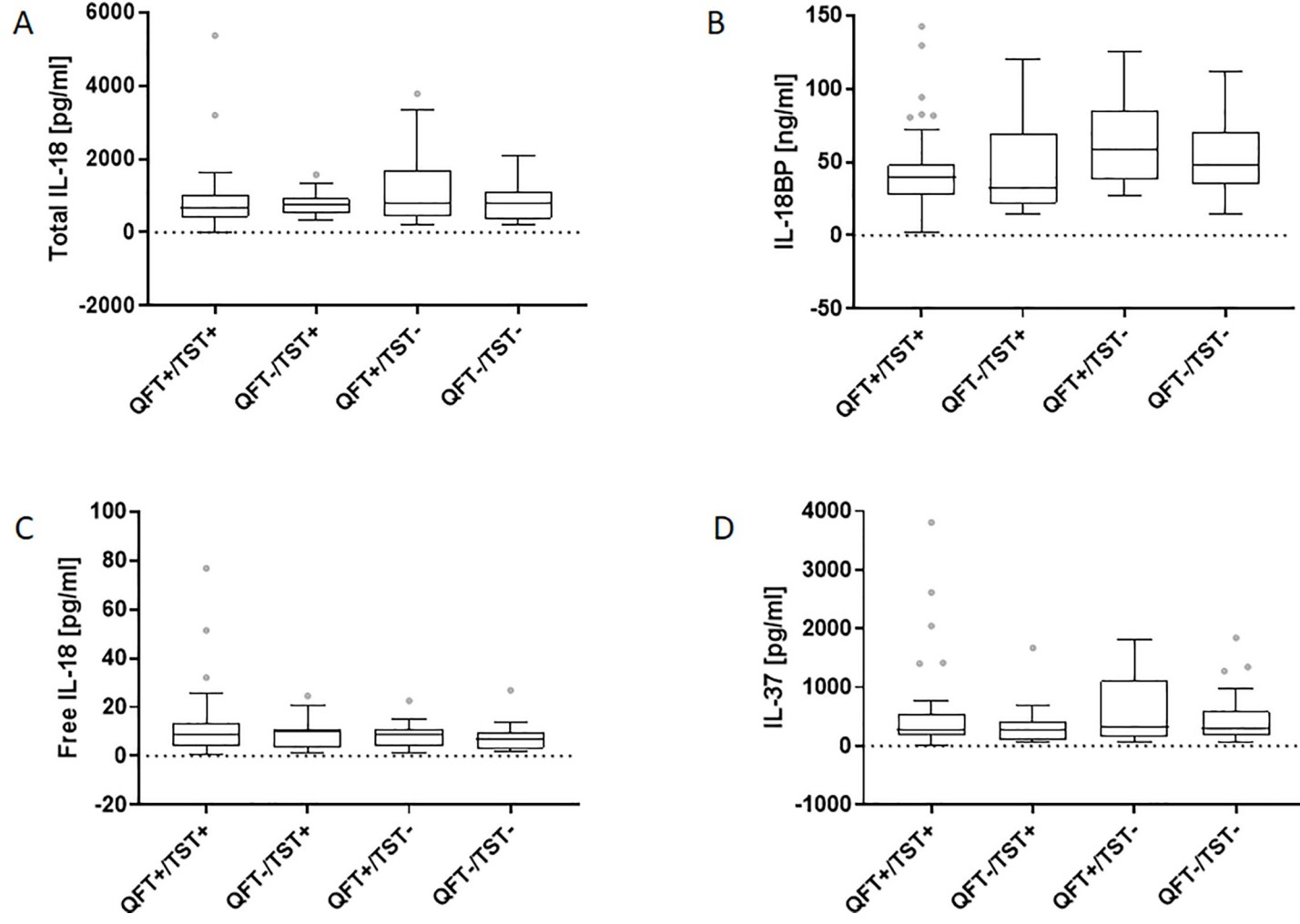

**Fig 4. Levels of total IL-18, IL-18BP, free IL-18 and IL-37 in *M.tb* antigens-stimulated QuantiFERON (QFT) supernatants versus QFT/TST result.** Boxplots with median (horizontal line within the box), interquartile range (box limits), and extremes (whiskers) of levels of total IL-18 (A), IL-18BP (B), free IL-18 (C), IL-37 (D) in *M. tb* antigens-stimulated QFT supernatants from active tuberculosis patients with positive or negative QuantiFERON (QFT) and tuberculin skin test (TST) results.

IL-18BP/IFN-γ and IL-37/IFN-γ protein ratios had the highest importance scores. The levels of IL-18, IL-18BP and IFN-γ and the ratios between IL-18/IL-37 and IL-37/IFN-γ remained informative in the joint analyses. IL-18 and IFN-γ were most informative for the LTBI vs. ATB comparison, and the IL-18/IL-18BP, IL-18/IL-37, IL-18/IFN-γ, IL-18BP/IL-37 and IL-18BP/

**Table 4. Predictive values (median AUC and 95% CI) of individual protein levels in QFT supernatants.**

| Parameter | HC vs. ATB | HC vs. LTBI | LTBI vs. ATB |
|---|---|---|---|
| | median AUC (95% CI) | | |
| IL-18 | 0.76 (0.76;0.77) | 0.51 (0.50;0.54) | 0.71 (0.69;0.72) |
| IL-18BP | 0.64 (0.64;0.65) | 0.48 (0.45;0.55) | 0.58 (0.56;0.59) |
| IL-37 | 0.54 (0.50;0.58) | 0.75 (0.69;0.83) | 0.59 (0.55;0.63) |
| IFN-γ | 0.83 (0.83;0.84) | 0.99 (0.99;0.99) | 0.58 (0.55;0.62) |

ATB–active tuberculosis (n = 95), HC- healthy controls (n = 91), LTBI–latent *M.tb* infection (n = 52)

**Table 5. Correlation (r) and p-values between protein levels in QFT supernatants.**

| Parameters | HC | | LTBI | | ATB | |
|---|---|---|---|---|---|---|
| | n = 91 | | n = 52 | | n = 95 | |
| | r | p | r | p | r | p |
| IL-18 ~ IL-18BP | -0.164 | 0.021 | -0.006 | 0.943 | 0.148 | 0.033 |
| IL-18 ~ IL-37 | -0.136 | 0.056 | -0.006 | 0.949 | -0.055 | 0.427 |
| IL-18 ~ IFN-γ | -0.139 | 0.051 | -0.075 | 0.430 | 0.011 | 0.867 |
| IL-18BP ~ IL-37 | 0.042 | 0.550 | -0.252 | 0.008 | -0.053 | 0.446 |
| IL-18BP ~ IFN-γ | 0.023 | 0.742 | -0.020 | 0.831 | 0.016 | 0.816 |
| IL-37 ~ IFN-γ | 0.257 | <0.001 | -0.092 | 0.335 | 0.023 | 0.733 |

ATB–active tuberculosis, HC- healthy controls, LTBI–latent *M.tb* infection

IFN-γ protein ratios had the highest importance scores. The levels of IL-18, IL-18BP and IFN-γ and the IL-18/IL-18BP, IL-18/IL-37, IL-18/IFN-γ and IL-18BP/IFN-γ protein ratios remained informative in the joint analyses.

IL-18 and IL-37 were the most informative serum markers for the LTBI vs. HC comparison. The protein ratios IL-18/IFN-γ, IL-18BP/IFN-γ, and IL-18BP/IP-10 had the highest importance scores, and the levels of IL-18, IL-18BP and IFN-γ and the IL-18/IFN-γ, IL-18BP/IL-37, IL-18BP/IFN-γ and IL-18BP/IP-10 protein ratios remained informative in the joint analyses. IL-18 and IP-10 were most informative for ATB vs. HC comparison, and all of the ratios, except IL-18/IP-10 and IL-37/IFN-γ, were informative. The levels of IL-18 and IP-10 had the highest discriminative power in the joint analysis. IL-18, IL-18BP and IP-10 were the most informative for the LTBI vs. ATB comparison, and the IL-18/IL-18BP, IL-18/IL-37, IL-18/IFN-γ and IL-18BP/IP-10 ratios had highest importance scores. The levels of IL-18, IL-18BP and IP-10 and the IL-18/IL-18BP, IL-18/IL-37, IL-18/IFN-γ and IL-18BP/IP-10 ratios remained informative in the joint analysis. The AUC values are summarized in Table 6.

The three-class comparison of the markers quantified in QFT supernatants revealed that all single protein levels were informative with a multi-class AUC = 0.7728. The IL-18/IFN-γ, IL-18BP/IFN-γ and IL-37/IFN-γ protein ratios had the highest importance scores with an AUC = 0.6026. IL-18 and IFN-γ and the IL-18/IFN-γ and IL-18BP/IFN-γ ratios were the most informative in the joint model with an AUC = 0.772.

Only IL-18 and IP-10 in the serum were informative with a multi-class AUC = 0.6507. All of the ratios, except IL-37/IFN-γ, were informative with an AUC = 0.6192. IL-18, IL-18BP and IP-10 as wells as the IL-18/IL-37, IL-18/IFN-γ and IL-18BP/IP-10 protein ratios were informative in the joint model with an AUC = 0.657.

**Table 6. AUC values for selected decision trees.**

| Comparison | QFT supernatant | | | Serum | | |
|---|---|---|---|---|---|---|
| | expression | ratio | both | expression | ratio | both |
| HC vs. LTBI | 0.997 | 0.940 | 0.976 | 0.561 | 0.501 | 0.502 |
| HC vs. ATB | 0.883 | 0.722 | 0.906 | 0.796 | 0.733 | 0.798 |
| LTBI vs. ATB | 0.751 | 0.722 | 0.773 | 0.758 | 0.661 | 0.739 |

ATB–active tuberculosis (n = 95), HC- healthy controls (n = 91), LTBI–latent *M.tb* infection (n = 52)

For each case, the first value (expression) refers to the model with levels of single proteins only, the second value (ratio) to the ratios between protein levels and the third value (both) to the joint model.

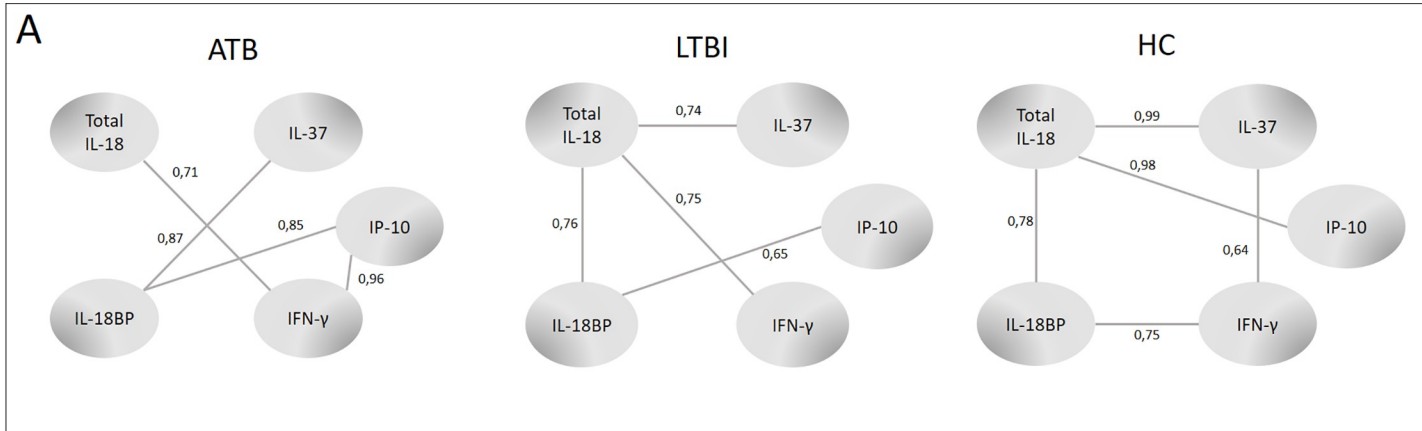

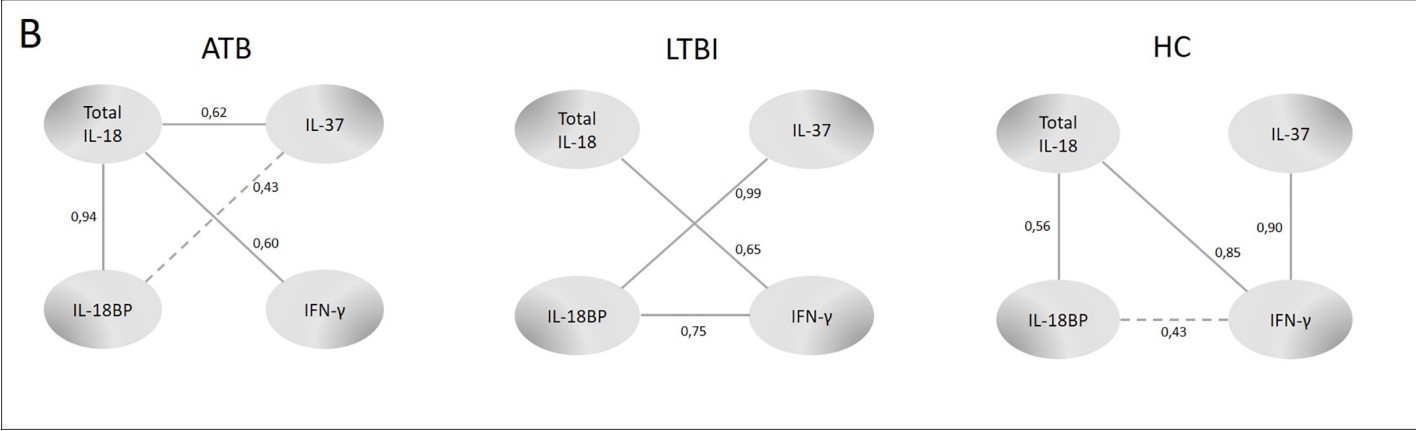

**Fig 5.** Network patterns among studied proteins measured in serum (A) and QFT supernatants (B) from pulmonary TB patients (ATB), latently infected subjects (LTBI) and non-infected healthy controls (HC) The numbers, all between 0 and 1, correspond to the 'likelihood' of observing a given co-expression in a network generated from a random subset of (at least 80% of) individuals in the group of interest.

## Variability of expression and co-expression levels

To shed some further light on the issue of insufficient discriminative power of single protein levels and protein ratios, we aim to perform a deeper study of the variability and co-expression of the selected markers. In addition to the differential expression between the study groups, IFN-γ levels had significantly different variance between HC and ATB and HC and LTBI in QFT supernatants ($p<10^{-10}$ in both cases). IL-18 was differentially variable between HC and LTBI in QFT supernatants ($p\sim10^{-4}$). IL-18, IL-18BP, IFN-γ and IP-10 in the serum samples were differentially variable between HC and LTBI, and IL-18 and IP-10 were differentially variable between LTBI and ATB.

We used the co-expression network-building method introduced by Hartmann et al. [21] and found that pairs of proteins were stably co-expressed in one study group and co-expressed in a varying fashion in another study group, which is a phenomenon that we call differential co-expression. The co-expression networks generated using this approach are presented in Fig 5. The relationships between (a) IP-10 and IFN-γ as well as between IL-18BP and IL-37 in the serum samples were most robust in the ATB group, (b) IL-18BP and IP-10 as well as IL-18 and IFN-γ were more robust in the ATB or LTBI groups than the HC group, and (c) IL-18 and IP-10, IL-18BP and IFN-γ as well as between IL-37 and IFN-γ were most robust in the HC group

**Table 7. Results for ordinal regression (elastic net)–coefficients.** All available predictors (5 levels of proteins and 10 ratios between these proteins in serum together with 4 protein levels and 6 ratios between proteins in QFT supernatants) were used to model the distribution of an ordinal random variable, Y, which equalled 1 for HC, 2 for LTBI and 3 for ATB. The non-zero coefficients are regarded as being informative of the (conditional) distribution of Y.

| Parameters | Coefficients (95% CI) | |
|---|---|---|
| | logit(P[Y = 1\|Y > = 1]) | logit(P[Y = 2\|Y > = 2]) |
| Serum | | |
| (Intercept) | 5.007 (2.320; 5.241) | 3.745 (2.144; 3.779) |
| IL-18 | -0.115 (-0.232; 0) | -0.073 (-0.232; 0) |
| IL-18BP | -0.237 (-0.311; -0.082) | -0.237 (-0.317; -0.093) |
| IL-37 | 0.129 (0; 0.047) | 0.046 (0; 0.002) |
| IFN-γ | -0.848 (-0.633; -0,020) | -0.848 (-0.633; -0.020) |
| IP-10 | -0.113 (-0.149; -0.062) | -0.157 (-0.175; -0.079) |
| IL-18/IL-18BP | -0.545 (-0.811; 0) | -1.025 (-0.858; 0) |
| IL-18/IL-37 | 0.129 (0;0.015) | 0.129 (0; 0.015) |
| IL-18/IFN-γ | -0.051 (-0.12; 0) | 0.079 (-0.007; 0.056) |
| IL-18/IP-10 | 0 (0; 0.154) | 0 (0; 0.088) |
| IL-18BP/IL-37 | 0 (-0.019; 0) | -0.153 (-0.081; 0) |
| IL-18BP/IFN-γ | 0 (-0.155; 0) | 0 (-0.209; 0) |
| IL-18BP/IP-10 | 0.222 (0.150; 0.431) | -0.005 (0; 0.430) |
| IL-37/IFN-γ | 0 (0; 0.023) | 0 (-0.007; 0.021) |
| IL-37/IP-10 | 0 (-0.034; 0) | -0.200 (-0.063; 0) |
| IFN-γ/IP-10 | 1.300 (0; 0.851) | 2.427 (0; 1.589) |
| QFT supernatant | | |
| IL-18 | -0.371 (-0.419; -0.063) | 0 (-0.123; 0) |
| IL-18BP | -0.097 (-0.540; -0.217) | -0.028 (-0.125; 0) |
| IL-37 | -0.086 (-0.067; 0) | 0.011 (-0.032; 0.004) |
| IFN-γ | -0.981 (-0.478; -0.153) | -0.027 (-0.027; 0) |
| IL-18/IL-18BP | 0.433 (0; 0) | -0.106 (-0.252; 0) |
| IL-18/IL-37 | 0 (-0.028; 0) | 0 (-0.035; 0) |
| IL-18/IFN-γ | 0.301 (0; 0.602) | -0.363 (-0.418; -0.021) |
| IL-18BP /IL-37 | -0.543 (-0.514; -0.074) | 0.089 (-0.022; 0.232) |
| IL-18BP/IFN-γ | 0.266 (0.411; 1. 116) | -0.659 (-0.947; -0.247) |
| IL-37/IFN-γ | 0.047 (0.031; 0.211) | -0.009 (0; 0.049) |

The CI's were estimated via boostrap procedure based on 300 repetitions with the size of the subsample equal to 80% of the original sample. This approach is proposed due to lack of closed form formulas for the distributions of the coefficients. Due to differences in sample size, there are instances where the coefficient lies outside of the CI indicating lack of robustness.

(Fig 5A). The relationship between (a) IL-18 and IL-37 in QFT supernatants was most robust in the ATB groups, (b) IL-18BP and IL-37 were more robust in the ATB or LTBI group than in the HC group, and (c) IL-37 and IFN-γ were most robust in the HC group (Fig 5B).

## Ordinal Elastic Net for the detection of protein levels and ratios predictive of the transition from HC through LTBI to ATB

The feature selection method based on ordinal regression with elastic net penalty revealed several important predictors in the levels and ratios of the sera and QFT supernatant proteins. All available predictors (5 levels of proteins and 10 ratios between these proteins in serum together with 4 protein levels and 6 ratios between proteins in QFT supernatants) were used to model

## Serum

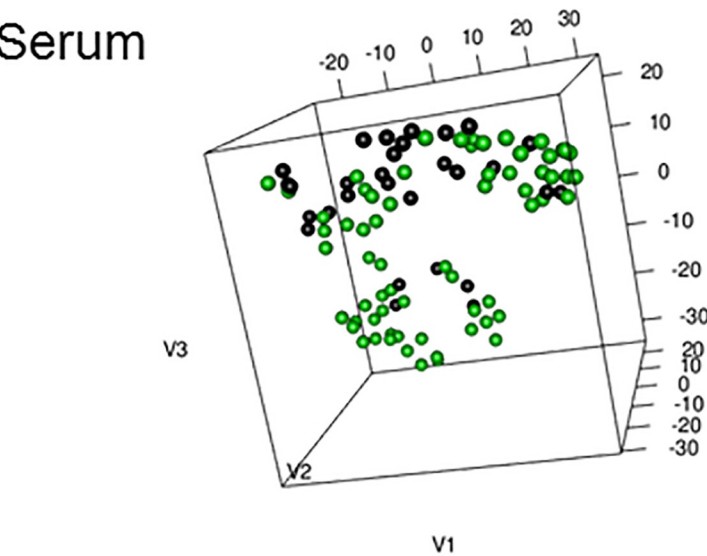

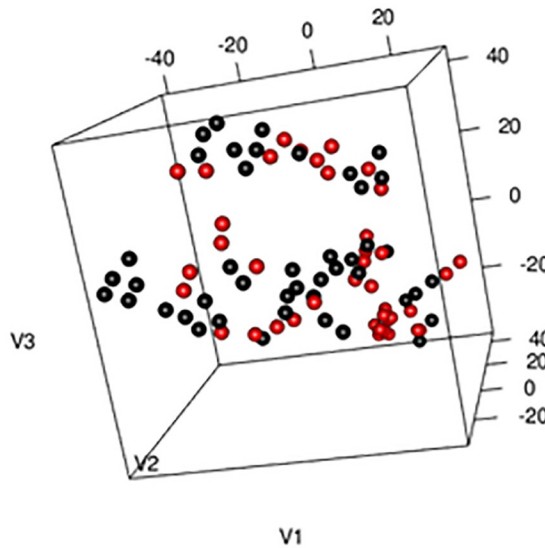

## QFT supernatant

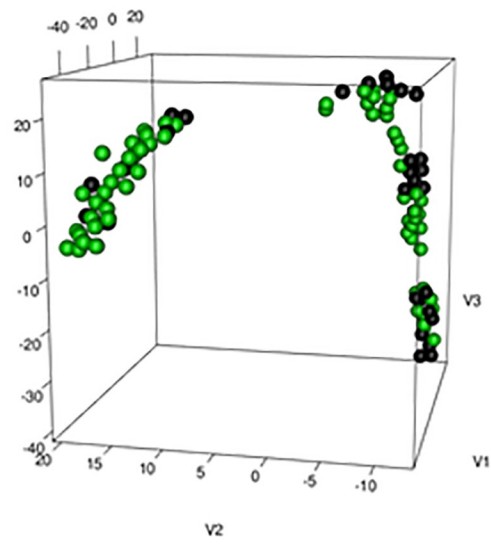

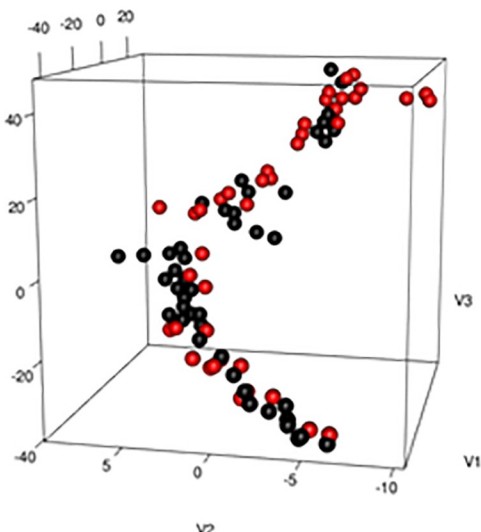

**Fig 6.** Results of the t-SNE algoritm on serum (top) and QFT supernatant (bottom). IGRA negative samples are in the left and IGRA positive in the right panel. ATB cases are in black, LTBI in red and HC in green.

the distribution of a random variable, Y, which equalled 1 for HC, 2 for LTBI and 3 for ATB. The performance of the model is presented in Table 7. Notably, the expression levels of all proteins in the serum and QFT supernatants were informative predictors of the infection status. However, a number of non-informative expression ratios were found: IL-18/IP-10, IL-18BP/IFN-γ, and IL-37/IFN-γ in the serum and IL-18/IL-37 in QFT supernatants.

### Unsupervised dimension reduction

We also asked whether the molecular signature based on the expression of the studied proteins and ratios may be used to define a meaningful partition of our cohort. We used an

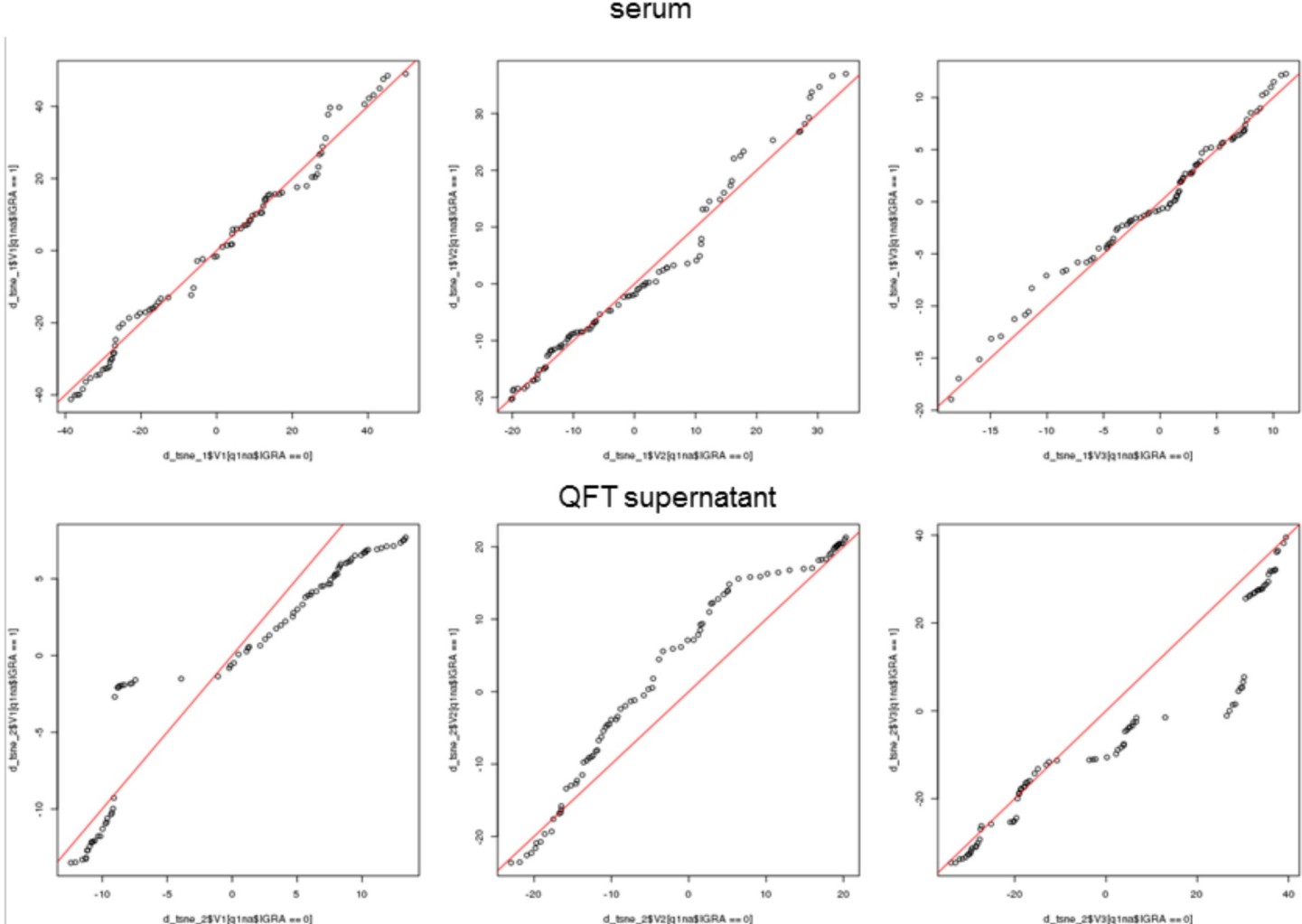

**Fig 7. qq-plots for the components of the t-SNE projection.** The top panel corresponds to the serum and the bottom to the QFT supernatant samples. The leftmost plot in each panel corresponds to the first coordinate of the embedding, the central plot to the second and the rightmost to the third coordinate of the embedding. The x-axis corresponds to IGRA negative and the y-axis to the IGRA positive samples. The red line is the diagonal. In the top panel (serum) the distribution of the consecutive components of the embedding is much more similar between IGRA positive and negative individuals, then in the bottom panel (QFT supernatants).

unsupervised dimension reduction technique, t-SNE, which is a non-linear alternative to standard Principal Component Analysis.

First, we performed t-SNE on serum and QFT supernatants separately, with no prior information on the studied individuals (Fig 6). For the serum samples, there was no clear clustering or separation between the studied groups. At the same time, there was a separation between the IGRA-negative and IGRA-positive samples for the QFT supernatants as confirmed in the analysis of the similarity of distributions of the three components (derived from t-SNE) between the IGRA-positive and -negative groups (see qq-plots in Fig 7). Then, we used the t-SNE conditionally on the IGRA result. In other words, we performed the dimension reduction separately for the IGRA-positive and -negative individuals. This prior information allowed us to identify two clusters for the IGRA-positive and IGRA-negative samples in the serum with a clearer separation between the ATB and HC groups than between the ATB and LTBI groups (only the active TB and HC could have a negative IGRA in our study) (Fig 6). For QFT

supernatant samples and IGRA-negative samples, we separated a group of healthy controls from the two clusters of the mixed samples (ATB and HC), and we noticed that the likelihood of the sample being classified as LTBI rather than ATB increased for IGRA-positive individuals with the increase in the value of the third projection of the embedding (Fig 6, Fig 7).

## Discussion

We estimated whether the levels of individual proteins of the IL-18 signalling complex i.e. total and free IL-18, IL-18BP, IL-37, IFN-γ and the IP-10 chemokine, and their mutual relationships and ratios, were useful as auxiliary biomarkers of ATB. The analyses showed a significant increase in the serum levels of total and free IL-18, IL-18BP, IFN-γ and IP-10 in ATB patients compared to LTBI or HC individuals. In contrast, a slightly lower serum concentration of the anti-inflammatory IL-37 was measured in ATB than in LTBI or HC groups [16]. This observation indicates a significant loss of balance in the range of the IL-18 signaling complex in ATB. An elevated serum IL-18 concentration in ATB had been previously demonstrated [22–25]. However, our study analysed the IL-18 signalling complex in a wider extent, by evaluating the total and free IL-18, IL-18BP, IL-37, IFN-γ and IP-10 chemokine, in two *M.tb* infection states, which to the best of our knowledge, had not been done previously. We performed statistical analyses of several classification algorithms to rank the measured proteins for their usefulness in the differentiation of *M.tb* infection states using unstimulated serum samples and *M.tb* antigen-stimulated QuantiFERON culture supernatants. Our data showed that individual serum proteins, except IL-37, were able to discriminate between ATB and LTBI in both groups of positive and negative QFT tests. However, the highest discriminative biomarker set was a complex co-expression of serum IL-18BP and IL-37 and IP-10 and IFN-γ, which may be useful in the rapid differentiation between ATB patients and LTBI individuals. Our data are consistent with the opinion that a complex biomarker panel is more robust than single markers for TB screening [5–7,26]. The set of seven serum biosignatures, comprised of apolipoprotein-A1, CRP, complement factor H, IFN-γ, IP-10, serum amyloid A and transthyretin, and a panel of five other serum biomarkers, including IFN-γ, IL-6, IL-18, CRP and MIG, showed potential in screening for TB in African countries endemic for HIV infection [8,27].

Our data suggest that the unstimulated biomarker performance is a better approach to evaluate the systemic manifestation of active TB compared to *M.tb*-stimulated marker expression. In contrast to the five individual serum proteins of the IL-18 signalling complex that were able to discriminate between the ATB and LTBI groups, only total IL-18 and IFN-γ showed significantly different values in QFT supernatants from ATB patients and LTBI individuals. Other results demonstrated that the cytokine ratios might provide specific and sensitive TB indicators [28–30]. A potential role of the IFN-γ/IL-2 ratio in the diagnosis of extrapulmonary TB was reported [31]. The IFN-γ/IL-4 and IL-4δ2/IL-4 mRNA ratios may serve as valuable markers for TB susceptibility or resistance [31,32]. La Manna et al. indicated that 14 analytes (IL-2, IP-10, IFN-γ, MIG, SCF, b-NGF, IL-12-p40, TRAIL, IL2Ra, MIF, TNF-β, IL-3, IFN-α 2, and LIF) allowed discriminating between ATB and non-TB groups [33].

The present study revealed significantly higher levels of circulating IFN-γ in ATB patients than in the LTBI or HC groups. In contrast, the IFN-γ concentration in *M.tb*-stimulated QFT cultures was lower in ATB compared to LTBI subjects. This difference suggests that the activation of an antimycobacterial immune response during ATB occurs concomitantly with the signs of immune depression [34]. However, it cannot be excluded that the most *M.tb*-reactive T cells are redistributed from the periphery to the site of infection, and consequently, only less responsive *M.tb*-specific T cells remain in the circulation. The elevated levels of serum IP-10 in ATB patients support this hypothesis because this chemokine recruits Th1 lymphocytes and

NK cells toward infected areas [35]. We previously reported that circulating leukocytes from healthy BCG-vaccinated individuals become effector cells producing IFN-γ upon stimulation with mycobacterial antigens [36]. The proportion of T CD4[+] Th1 cells synthesizing IFN-γ in these cultures significantly exceeded the proportion of T CD8[+] cells and NK cells. The IL-18 enhanced IFN-γ production by naïve rather than memory CD4[+] Th1 cells [37]. To perform this function, CD4[+] Th1 cells recognized *M.tb* antigens that were presented via major histocompatibility complex (MHC) class II molecules at the surface of dendritic cells. It is possible that antigen presentation to CD4[+] Th1 cells in active TB might not be optimal for the IFN-γ response in QFT cultures. With this in mind, the search for novel antigens, other than those used in IFN-γ-release assays (ESAT-6/CFP-10/TB7.7), was undertaken by Chegou et al. [38]. Alternatively, we also speculate that new cell culture models that mimic the microenvironment of human lung tissue [39] may be used for the development of a robust immunodiagnosis of TB. In regards to such a supposition, the co-expression of the proteins of the IL-18/IL37 signalling complex and IP-10 should be further analysed in patients with pulmonary disease including those with pulmonary disease other than TB.

## Conclusion

Our results show that the IL-18 signalling complex may be exploited by *M. tuberculosis* to expand the clinical manifestation of pulmonary TB. Therefore, direct analysis of serum components of the IL-18/IL-37 signalling complex and IP-10 may be applicable in designing novel rapid screening tests for pulmonary TB.

## Supporting information

**S1 File. Raw data serum.**
(XLSX)

**S2 File. Raw data QFT.**
(XLSX)

## Acknowledgments

This work was supported by the National Science Centre grants no 2015/19/N/NZ6/01385 and 2016/21/B/NZ7/01771.

## Author Contributions

**Conceptualization:** Sebastian Wawrocki, Magdalena Druszczynska.

**Data curation:** Sebastian Wawrocki, Grzegorz Kielnierowski.

**Formal analysis:** Magdalena Druszczynska.

**Funding acquisition:** Sebastian Wawrocki, Magdalena Druszczynska.

**Investigation:** Sebastian Wawrocki.

**Methodology:** Sebastian Wawrocki, Michal Seweryn.

**Project administration:** Magdalena Druszczynska.

**Resources:** Magdalena Druszczynska.

**Supervision:** Grzegorz Kielnierowski, Wieslawa Rudnicka, Magdalena Druszczynska.

**Validation:** Michal Seweryn.

**Visualization:** Sebastian Wawrocki, Michal Seweryn, Marcin Wlodarczyk.

**Writing – original draft:** Sebastian Wawrocki, Michal Seweryn, Wieslawa Rudnicka, Magdalena Druszczynska.

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
