## [Decision Letter · Decision Letter 0]

16 Jul 2019

PONE-D-19-15207

IL-18/IL-37/IP-10 signaling complex as a potential biomarker for discriminating active and latent TB

PLOS ONE

Dear Dr Druszczynska,

Thank you for submitting your manuscript to PLOS ONE. After careful consideration, we feel that it has merit but does not fully meet PLOS ONE’s publication criteria as it currently stands. Therefore, we invite you to submit a revised version of the manuscript that addresses the points raised during the review process.

We would appreciate receiving your revised manuscript by 60 days. To enhance the reproducibility of your results, we recommend that if applicable you deposit your laboratory protocols in protocols.io, where a protocol can be assigned its own identifier (DOI) such that it can be cited independently in the future. For instructions see: http://journals.plos.org/plosone/s/submission-guidelines#loc-laboratory-protocols

We look forward to receiving your revised manuscript.

Kind regards,

Francesco Dieli, M.D., Ph.D.

Academic Editor

PLOS ONE

Journal Requirements:

2. Please include in your Methods section the date ranges over which you recruited participants to this study.

Reviewers' comments:

Reviewer's Responses to Questions

**Comments to the Author**

1. Is the manuscript technically sound, and do the data support the conclusions?

Reviewer #1: Partly

Reviewer #2: Yes

Reviewer #3: Yes

2. Has the statistical analysis been performed appropriately and rigorously? 

Reviewer #1: I Don't Know

Reviewer #2: Yes

Reviewer #3: Yes

3. Have the authors made all data underlying the findings in their manuscript fully available?

Reviewer #1: No

Reviewer #2: Yes

Reviewer #3: No

4. Is the manuscript presented in an intelligible fashion and written in standard English?

Reviewer #1: No

Reviewer #2: Yes

Reviewer #3: Yes

5. Review Comments to the Author

Reviewer #1: In this paper the authors analyse the levels of IL18 signalling complex (IL18, IL18 BP, IL37 and free IL18) and the levels of IFNγ and IP10 in sera and in QFT plasma of Active TB patients (ATB), latent TB infected subjects (LTBI) and healthy controls (HC). They study, using several statistical tools, the better combination of Cytokines concentrations or ratio able to distinguish ATB from LTBI.

Even if the rationale of this study is interesting, I observed a certain number of critical points.

The authors do not specify where the original data are available

Introduction

On the contrary of what written at line 95, the authors measured the levels of IP10 only in serum and not in QFT plasma.

Methods

It is not clear if the authors in the analysis of QFT supernatants calculated the levels of the studied cytokines after subtraction of baseline levels obtained from NIL tube.

Results

Paragraph 2.2 and 2.3.

Line 203. About the ROC analysis of cytokines levels paragraph 2.2 the authors do not explain how is performed the full logistic model analysis. For none of the studied parameters they provide any threshold level with values of specificity and sensibility, the results are expressed in only AUC.

Line 225 the authors write that no significant differences were observed for the levels of IL18, IL18 BP, IL37 and free IL18 among the groups in QFT supernatants but in figure 3B is highlited a significant difference of the IL18BP levels between ATB and HC.

Line 226 the analysis of IFN-γ among IGRA+ and IGRA- and TST+ and TST-, in my opinion it should be deeply revised or eliminated because it is quite obvious that a comparison of IFN-γ levels between QFT+ and QFT- give a significant difference, these data are consistent with the algorithm used in QFT TB GOLD in tube test, so they do not add any original information (fig 4E).

Line 228 the values of IFN-γ reported in this line do not fit with those showed in the Figure 3E for the group of HC which in the line is reported as “culture negative” I suppose.

Paragraph 2.4

Line 265 the authors write that IL18, IL18BP and IFN-g from QFT supernatants are the most informative but at line 225 they write that there was no significant difference among the groups about the levels of IL18 and IL18BP.

Paragraph 2.5

The results of this work are based on the analysis of the levels of cytokines studied using different statistical tests. In this way the authors, carrying out a meticulous statistical analysis, generate a considerable quantity of results. the exposure of the results, in some cases, seems to me rather confusing. It would be useful especially for the paragraph 2.5, a table that summarizes the values of the different ratios studied and not only the table with the AUC obtained from the combined analyses of the levels of cytokines and ratio.

About the tables in which the authors show the AUC resulted from the analysis of the levels of single cytokine in serum or in QFT supernatants, in my opinion they should also show the confidence interval for each AUC and for the parameters that show highest discrimination power among the groups they should propose a threshold value with relative sensibility and specificity.

Conclusion

The author suggests that the analysis of IL18 Signalling complex in serum may be applicable for a rapid screening test for pulmonary TB. In my opinion it would be necessary to add another group of patients, those with pulmonary disease other than TB. The levels of Cytokine observed in serum are baseline levels and not obtained with specific antigen stimulation so they reflect an inflammatory status rather than a cytokine pattern caused by a specific antigenic response, as that observed in QFT supernatants. This inflammatory status could be observed also in other pulmonary diseases, for this reason the results obtained from serum analysis in any case should be coupled with an IGRA test to be sure that that the subject is infected with Mtb.

Reviewer #2: The study suggest the evaluation of IL8 signalling to design a novel screening test for pulmonary TB. In the discussion the authors should argument also the application of immunological biomarkers as correlate of protection or disease: it could be interesting to study the modulation of IL8 signalling in LTBI and active TB population over time-therapy, in order to find an eventually find a correlation between the resolution of the disease and the cytokine level. Moreover it could be interesting to monitor the IL8 signaling in a latent population not taking preventive therapy, with the aim to find biomarkers of incipient active TB disease. Identify the LTBI subjects with the higher risk to develop active TB is currently one of the most important research topic in the TB field. In high TB endemic country it is not possible to offer TB preventive therapy to all LTBI subjects. Therefore, to find correlates of disease or protection is a way to offer TB preventive therapy to a selected population. Considering the impossibility to isolate Mtb in LTBI population, the use of immunological biomarkers is an alternative and promising strategy.

The study is well augmented but need some minor revision. In some part of the text, the description of figures and table is to poor.

• In the table 1 should be added:

o the origin of the enrolled patients

o appropriate statistical analysis among groups

o results about microbiological and clinical diagnosis of TB patients reported in the text

• line 132 correct QFT

• line 184, specify that these data are not reported in the figure

• line 200 the labels of the figure in not corrected, it should be 2 A,D,F

• figure 2B, 2C,2E and 2D are not discussed in the text

• line 203, in order to better understand the results, the data related to the full logistic model should be included in the table 2

• line 224-226 : in the text it is reported “There were no significant differences in the levels of free IL-18, IL-18BP or IL-37 in the QFT 226 supernatants among studied groups (Fig. 3B-D)”. In the figure 3B it is reported a significant differences comparing active TB with HC. Please clarify

• 229-230 “ IFN-g level was significantly higher in than in QFT-/TST+/TST- ATB patients (Fig. 4E).” “QFT-/TST+/TST-“ is not the correct definition of the group reported in the figure 4. Please clarify.

• In the figure 4 is not reported the legend, it is not clear what circle and triangle represents.

• The figure legend of all manuscript should be improved with a brief description of the data, for instance: serum, QFT plasma, analyzed groups, statistic used, graph legend.

• line 238, in order to better understand the results, the data related to the full logistic model should be included in the table 4

• line 353 : please specify that only the active TB and HC could have a negative IGRA.

• Regarding the table 3 I did not understand the correspondent comment reported in the text: Line 221 you wrote that the pair wise relationships is independent of the Mtb infection for IL18BP and IP 10 but in the table it is reported a significant relationship in LTBI and active TB. Please clarify what you mean. I found similar discrepancy also in the other description, please describe better this part.

• Regarding the table 7, the author should describe better and clearer what it reported in the table.

Reviewer #3: From my view, the first issue is only notational, but it must be fixed. Methodologically, my only concern is the skewness of the distributions. According to this, the authors apply non-parametric methods but they also use some indexes for symmetrical distributions as I will comment below.

Regarding notation:

I guess that BCG stands for Bacillus Calmette-Guerin and TST stands for Tuberculin Skin Test, but these acronyms are not properly introduced in the manuscript

According to usual statistical guidelines (e.g. APA), p values should be reported with 3 figures, unless they are lower than .001, in that case, it is enough reporting p<.001. It is senseless a p-value with 7 figures (line 179 and Fig. 1A). If the authors consider relevant to report p-values with higher accuracy (my view is that this is not the case), they must consider exponential notation. There are more p-values reported with an excess of decimal figures (lines 187, 224, tables 3, 5, figure 3, …) and also with few decimals (lines 190, 191, 194, …). Please, fix this issue.

This is just a comment, not an issue: I wander what is the interest of reporting AUC with four decimals.

Standard deviation must have the same accuracy or one decimal figure more than the mean value. Please, fix the expression in line 194.

In tables 2 and 4, change the decimal point to a point.

This is just another comment, using “±” to indicate mean and the standard deviation is increasingly discouraged in most of the guidelines for statistical reporting.

I miss the SE of the AUC estimates

Regarding methodology:

Figures 1-4 show that the IL distributions seems to be quite skewed. In fact, the authors have considered (properly) a non-parametric K-S test in order to check the homogeneity between groups. But they report the descriptive measures as mean and standard deviations. This can be misleading. For example, IL-8 in figure A has a variation coefficient greater than 100% in two groups. My view is that the descriptive statistics should be given in the same vein (eg median and IQR). An alternative choice is to transform data in order to induce symmetry.

This issue also affects the choice of the Pearson correlation coefficient in order to assess correlations. Pearson coefficient can be very sensitive to the presence of outliers. I wonder why the authors have not considered a non-parametric correlation coefficient, as Spearman's rho.

My view is that coefficients in fig 5, and the quartile plots in fig 7, should have some diagnostic words to help the reader to interpret them.

Line 430, please, change “demonstrate” by show, illustrate, evidence ...

6. PLOS authors have the option to publish the peer review history of their article (what does this mean?). If published, this will include your full peer review and any attached files.

Reviewer #1: No

Reviewer #2: No

Reviewer #3: Yes: Pedro Femia

---

## [Author Response · Author response to Decision Letter 0]

20 Sep 2019

To Reviewer #1: 

We would like to thank you kindly for reading our manuscript “IL-18/IL-37/IP-10 signaling complex as a potential biomarker for discriminating active and latent TB” and for all the questions and suggestions. 

Our answers to the questions are as follows:

1. „The authors do not specify where the original data are available.”

All the data are available in the Supporting information file.

2. „Introduction. On the contrary of what written at line 95, the authors measured the levels of IP10 only in serum and not in QFT plasma.”

The sentence was corrected (p.4-5, lines 96-101).

3. „Methods. It is not clear if the authors in the analysis of QFT supernatants calculated the levels of the studied cytokines after subtraction of baseline levels obtained from NIL tube.”

The levels of the studied cytokines were calculated after subtraction of baseline levels obtained from NIL tube. The appropriate sentence was included in the text (p. 7, lines 161-162).

4. „Results. Paragraph 2.2 and 2.3. Line 203. About the ROC analysis of cytokines levels paragraph 2.2 the authors do not explain how is performed the full logistic model analysis. For none of the studied parameters they provide any threshold level with values of specificity and sensibility, the results are expressed in only AUC.”

The appropriate text was added in the Methods section: “For the classical analysis of association between levels of expression of single proteins and protein ratios logistic regression was used in order to make the comparison of performance of single markers to all available markers (full logistic model) 'fair'. The AUC values were estimated via 5-fold cross-validation and based on at least 500 bootstrap replicates” (p.8 , lines 171-175). The 95% CI for the AUC was added to the appropriate tables (Table 2 and 4) and to the text (p. 10, lines 232-235, p. 12, lines 285-287, p.13, lines 307-312)

5. „Line 225 the authors write that no significant differences were observed for the levels of IL18, IL18 BP, IL37 and free IL18 among the groups in QFT supernatants but in figure 3B is highlited a significant difference of the IL18BP levels between ATB and HC.”

The appropriate sentences were corrected (p. 11, lines 256-259).

6. „Line 226 the analysis of IFN-γ among IGRA+ and IGRA- and TST+ and TST-, in my opinion it should be deeply revised or eliminated because it is quite obvious that a comparison of IFN-γ levels between QFT+ and QFT- give a significant difference, these data are consistent with the algorithm used in QFT TB GOLD in tube test, so they do not add any original information (fig 4E).”

The figure 4E and its description in the text were removed.

7. „Line 228 the values of IFN-γ reported in this line do not fit with those showed in the Figure 3E for the group of HC which in the line is reported as “culture negative” I suppose.”

The sentence was modified (p. 11-12, lines 260-263).

8. „Paragraph 2.4. Line 265 the authors write that IL18, IL18BP and IFN-g from QFT supernatants are the most informative but at line 225 they write that there was no significant difference among the groups about the levels of IL18 and IL18BP.”

The sentence was adjusted: “Even with no statistically significant differences between studied groups, the IL-18, IL-18BP and IFN-γ from QFT supernatants were most informative (in terms of AUC) in the LTBI vs. HC comparison” (p. 14, lines 318-320)

9. „Paragraph 2.5. The results of this work are based on the analysis of the levels of cytokines studied using different statistical tests. In this way the authors, carrying out a meticulous statistical analysis, generate a considerable quantity of results. the exposure of the results, in some cases, seems to me rather confusing. It would be useful especially for the paragraph 2.5, a table that summarizes the values of the different ratios studied and not only the table with the AUC obtained from the combined analyses of the levels of cytokines and ratio.”

The aim of paragraph 2.5 was not to evaluate the performance (in terms of AUC) of the selected proteins and protein ratios, but to shed some light on the poor performance of some of these features in terms of discriminative power. One of the reasons we highlight is differential variability and differential co-expression, which affects both the results of significance tests as well as predictive power. A sentence at the beginning of 2.5 was added: “To shed some further light on the issue of insufficient discriminative power of single protein levels and protein ratios, we aim to perform a deeper study of the variability and co-expression of the selected markers.” (p. 16, lines 356-358).

10. „About the tables in which the authors show the AUC resulted from the analysis of the levels of single cytokine in serum or in QFT supernatants, in my opinion they should also show the confidence interval for each AUC and for the parameters that show highest discrimination power among the groups they should propose a threshold value with relative sensibility and specificity.”

The CI for the logistic models was added. The cutoff value we choose is based on bootstrap replicates, thus we cannot guarantee (due to the lack of appropriate model/distribution for the cutoff value) that the median (or mean) cutoff will be in any sense optimal. We feel that the analysis of sensitivity and specificity of the proposed tests (and the relative trade-off between these two) goes beyond the scope of this short article. We enclose tables of both sensitivity and specificity for the Reviewer convenience. 

Serum

QFT supernatant

11. „Conclusion. The author suggests that the analysis of IL18 Signalling complex in serum may be applicable for a rapid screening test for pulmonary TB. In my opinion it would be necessary to add another group of patients, those with pulmonary disease other than TB. The levels of Cytokine observed in serum are baseline levels and not obtained with specific antigen stimulation so they reflect an inflammatory rather than a cytokine pattern caused by a specific antigenic response, as that observed in QFT supernatants. This inflammatory status could be observed also in other pulmonary diseases, for this reason the results obtained from serum analysis in any case should be coupled with an IGRA test to be sure that that the subject is infected with Mtb.”

We agree that the concentrations of the proteins of IL-18 signalling complex should be investigated in patients with nonmycobacterial pulmonary diseases and in the future studies, we are planning to analyze the levels of studied immunomarkers in patients with pulmonary disease including those with pulmonary disease other than TB. We introduced such sentence in the Discussion section of the manuscript (p.21, lines 497-500). 

In our study, 37% of patients with active TB showed negative IGRA test, Table 1.

To Reviewer #2

We would like to thank you kindly for reading our manuscript “IL-18/IL-37/IP-10 signaling complex as a potential biomarker for discriminating active and latent TB” and for all the questions and suggestions. 

Our answers to the questions are as follows:

1.„The study suggest the evaluation of IL8 signalling to design a novel screening test for pulmonary TB. In the discussion the authors should argument also the application of immunological biomarkers as correlate of protection or disease: it could be interesting to study the modulation of IL8 signalling in LTBI and active TB population over time-therapy, in order to find an eventually find a correlation between the resolution of the disease and the cytokine level. Moreover it could be interesting to monitor the IL8 signaling in a latent population not taking preventive therapy, with the aim to find biomarkers of incipient active TB disease. Identify the LTBI subjects with the higher risk to develop active TB is currently one of the most important research topic in the TB field. In high TB endemic country it is not possible to offer TB preventive therapy to all LTBI subjects. Therefore, to find correlates of disease or protection is a way to offer TB preventive therapy to a selected population. Considering the impossibility to isolate Mtb in LTBI population, the use of immunological biomarkers is an alternative and promising strategy.”

We agree that the identification of M.tb-infected individuals and the appropriate treatment of people with the higher TB development risk are undoubtedly crucial for effective TB control (Introduction section, p. 3, lines 65-72).In our many years of research, we found that about 13% of adult Poles are latently infected with M.tb [Druszczynska et al, Clin Dev Immunol 2013, doi: 10.1155/2013/851452]. It is impossible to embrace such a large group of people with preventive therapy.

2. „In some part of the text, the description of figures and table is to poor.”

The description of figures and tables was revised.

3. „ In the table 1 should be added:

o the origin of the enrolled patients

o appropriate statistical analysis among groups

o results about microbiological and clinical diagnosis of TB patients reported in the text”

Table 1 was modified and all suggested information was included.

4.”• line 132 correct QFT”

The abbreviation was corrected (p. 7, line 152).

5”• line 184, specify that these data are not reported in the figure”

The information is provided in brackets (p. 9, lines 200-201).

6”• line 200 the labels of the figure in not corrected, it should be 2 A,D,F

• figure 2B, 2C,2E and 2D are not discussed in the text”

The sentence was modified and all the figures were discussed in the text (p. 10, lines 220-222)

7”• line 203, in order to better understand the results, the data related to the full logistic model should be included in the table 2”

The performance of the full logistic model is included in the main text. The CIs for the AUC were added.

8”• line 224-226 : in the text it is reported “There were no significant differences in the levels of free IL-18, IL-18BP or IL-37 in the QFT 226 supernatants among studied groups (Fig. 3B-D)”. In the figure 3B it is reported a significant differences comparing active TB with HC. Please clarify”

The appropriate sentences were corrected (p. 11, lines 256-259).

9”• 229-230 “ IFN-g level was significantly higher in than in QFT-/TST+/TST- ATB patients (Fig. 4E).” “QFT-/TST+/TST-“ is not the correct definition of the group reported in the figure 4. Please clarify.”

We agree that it was not the correct definition of the group reported in the Fig. 4.The sentence was removed from the text as suggested by the Reviewer 1. 

10”• In the figure 4 is not reported the legend, it is not clear what circle and triangle represents.”

The figure was changed.

11”• The figure legend of all manuscript should be improved with a brief description of the data, for instance: serum, QFT plasma, analyzed groups, statistic used, graph legend.”

The description of figures was revised.

12”• line 238, in order to better understand the results, the data related to the full logistic model should be included in the table 4”

The performance of the full logistic model is included in the main text. The CIs for the AUC were added.

13”• line 353 : please specify that only the active TB and HC could have a negative IGRA”.

The information was added (p. 18, lines 421-422).

14”• Regarding the table 3 I did not understand the correspondent comment reported in the text: Line 221 you wrote that the pair wise relationships is independent of the Mtb infection for IL18BP and IP 10 but in the table it is reported a significant relationship in LTBI and active TB. Please clarify what you mean. I found similar discrepancy also in the other description, please describe better this part.”

The corresponding text was modified (p.10-11, lines 241-245).

15”• Regarding the table 7, the author should describe better and clearer what it reported in the table”.

The following text was added: “All available predictors (5 levels of proteins and 10 ratios between these proteins in serum together with 4 protein levels and 6 ratios between proteins in QFT supernatants) were used to model the distribution of an ordinal random variable, Y, which equalled 1 for HC, 2 for LTBI and 3 for ATB. The non-zero coefficients are regarded as being informative of the (conditional) distribution of Y.”

To Reviewer #3:

We would like to thank you kindly for reading our manuscript “IL-18/IL-37/IP-10 signaling complex as a potential biomarker for discriminating active and latent TB” and for all the questions and suggestions. 

Our answers to the questions are as follows:

1. „Regarding notation: I guess that BCG stands for Bacillus Calmette-Guerin and TST stands for Tuberculin Skin Test, but these acronyms are not properly introduced in the manuscript.”

The abbreviations were explained in the sentences where they first appeared (p. 5, lines 105 and 120).

2. „According to usual statistical guidelines (e.g. APA), p values should be reported with 3 figures, unless they are lower than .001, in that case, it is enough reporting p<.001. It is senseless a p-value with 7 figures (line 179 and Fig. 1A). If the authors consider relevant to report p-values with higher accuracy (my view is that this is not the case), they must consider exponential notation. There are more p-values reported with an excess of decimal figures (lines 187, 224, tables 3, 5, figure 3, …) and also with few decimals (lines 190, 191, 194, …). Please, fix this issue.”

We reported the p values as p<.001 when they were lower than 0.001. We fixed also all the values in lines, tables and figures with an excess or shortage of the decimal places as suggested. 

3. „This is just a comment, not an issue: I wander what is the interest of reporting AUC with four decimals.”

We apologize for the inconvenience, the numbers in tables are now corrected. We reported AUC up to four decimals as the CIs for the AUC were rather narrow in some of the studied cases. Additionally, as the estimates were based on the bootstrap, the four decimals (with narrow CIs) allowed to detect cases in which the detailed investigation of whether the bootstrap does not produce heavily biased estimates was needed.

4. „Standard deviation must have the same accuracy or one decimal figure more than the mean value. Please, fix the expression in line 194.”

We fixed all the values in lines, tables and figures with an excess or shortage of the decimal places as suggested. 

5. „In tables 2 and 4, change the decimal point to a point.”

It was corrected.

6. „This is just another comment, using “±” to indicate mean and the standard deviation is increasingly discouraged in most of the guidelines for statistical reporting.”

We agree. We modified Fig 1-4 as the Reviewer suggested showing median and IQR instead of mean and standard deviation. We modified also the corresponding text in the Results section (p. 9-10, lines 199-220, p. 11-12, lines 252-263). 

7. „I miss the SE of the AUC estimates”

95% CIs were added. 

8. „Regarding methodology: Figures 1-4 show that the IL distributions seems to be quite skewed. In fact, the authors have considered (properly) a non-parametric K-S test in order to check the homogeneity between groups. But they report the descriptive measures as mean and standard deviations. This can be misleading. For example, IL-8 in figure A has a variation coefficient greater than 100% in two groups. My view is that the descriptive statistics should be given in the same vein (eg median and IQR). An alternative choice is to transform data in order to induce symmetry.”

We modified Fig 1-4 as the Reviewer suggested showing median and IQR instead of mean and standard deviation. We modified also the corresponding text in the Results section (p. 9 lines 197-218, p. 11 lines 246-257).

9. „This issue also affects the choice of the Pearson correlation coefficient in order to assess correlations. Pearson coefficient can be very sensitive to the presence of outliers. I wonder why the authors have not considered a non-parametric correlation coefficient, as Spearman's rho”.

We investigated the relation by two other commonly used rank-based association measures (Spearman's rho and Kendall's tau) – both of these gave consistent results, therefore, we used the simplest estimate.

10. „My view is that coefficients in fig 5, and the quartile plots in fig 7, should have some diagnostic words to help the reader to interpret them.”

For Fig 5. the following text was added: “The numbers, all between 0 and 1, correspond to the 'likelihood' of observing a given co-expression in a network generated from a random subset of (at least 80% of) individuals in the group of interest.”

Also, for Fig 7. the following text was added: “In the top panel (serum) the distribution of the consecutive components of the embedding is much more similar between IGRA positive and negative individuals, then in the bottom panel (QFT supernatants).”

11. „Line 430, please, change “demonstrate” by show, illustrate, evidence ...”

 We changed the verb „demonstrate” to „show” (p. 22, line 503).

---

## [Decision Letter · Decision Letter 1]

11 Oct 2019

PONE-D-19-15207R1

IL-18/IL-37/IP-10 signaling complex as a potential biomarker for discriminating active and latent TB

PLOS ONE

Dear Dr Druszczynska,

Thank you for submitting your manuscript to PLOS ONE. After careful consideration, we feel that it has merit but does not fully meet PLOS ONE’s publication criteria as it currently stands. Therefore, we invite you to submit a revised version of the manuscript that addresses the points raised during the review process.

ACADEMIC EDITOR: Please address all the issues raised by the reviewers on the manuscript, not just on the rebuttal letter.

We would appreciate receiving your revised manuscript by Nov 25 2019 11:59PM. To enhance the reproducibility of your results, we recommend that if applicable you deposit your laboratory protocols in protocols.io, where a protocol can be assigned its own identifier (DOI) such that it can be cited independently in the future. For instructions see: http://journals.plos.org/plosone/s/submission-guidelines#loc-laboratory-protocols

We look forward to receiving your revised manuscript.

Kind regards,

Selvakumar Subbian, Ph.D.

Academic Editor

PLOS ONE

Reviewers' comments:

Reviewer's Responses to Questions

**Comments to the Author**

1. If the authors have adequately addressed your comments raised in a previous round of review and you feel that this manuscript is now acceptable for publication, you may indicate that here to bypass the “Comments to the Author” section, enter your conflict of interest statement in the “Confidential to Editor” section, and submit your "Accept" recommendation.

Reviewer #1: All comments have been addressed

Reviewer #3: (No Response)

2. Is the manuscript technically sound, and do the data support the conclusions?

Reviewer #1: Yes

Reviewer #3: Yes

3. Has the statistical analysis been performed appropriately and rigorously? 

Reviewer #1: Yes

Reviewer #3: Yes

4. Have the authors made all data underlying the findings in their manuscript fully available?

Reviewer #1: Yes

Reviewer #3: Yes

5. Is the manuscript presented in an intelligible fashion and written in standard English?

Reviewer #1: Yes

Reviewer #3: (No Response)

6. Review Comments to the Author

Reviewer #1: The authors have improved the paper according to almost all the suggestions of reviewers. The paper is suitable for publication after some minor revision.

Line 130 Fifty-two % should be replaced by fifty-five% according to the number in brackets.

Line 187 I assume that "Me" is the abbreviation for median but it was not introduced previously in the manuscript.

Line 210 I suppose that average should be replaced by median.

Line 285 please write also the CI of AUC for HC vs ATB.

Line 365 and fig 5 the group ATB in the manuscript is written TB in the figure 5, could you please replace TB with ATB in the figure.

Line 414: in the manuscript the authors mentioned “squares” as symbol for IGRA negative patients in figure 6, but I don’t see squares in figure 6, only circles of different colours. The authors should fix this point

Lines 488, 489, 490 and 493 please write CD4+ with the symbol + superscript.

Reviewer #3: My view is that this manuscript still has several minor issues and it needs some review from the authors.

Regarding decimals homogeneity

- P-values in the text (lines 209 to 215, p-values are still shown with two -rather than three- decimal places)

- CI values in table 4 (IFN-gamma/HC vs LTBI CI)

- Table 6 (decimals and n)

Tables (e.g. tables 3 and 5) should show always the sample sizes involved in comparisons and correlations, regardless this information has already given in the text

Table 7: please show the SE of the coefficients or their CI

7. PLOS authors have the option to publish the peer review history of their article (what does this mean?). If published, this will include your full peer review and any attached files.

Reviewer #1: No

Reviewer #3: No

---

## [Author Response · Author response to Decision Letter 1]

5 Nov 2019

To Reviewer #1: 

We would like to thank you kindly for reading our manuscript “IL-18/IL-37/IP-10 signaling complex as a potential biomarker for discriminating active and latent TB” and for all the questions and suggestions. 

Our answers to the questions are as follows:

1. Line 130 Fifty-two % should be replaced by fifty-five% according to the number in brackets.

We modified the sentence as suggested (p.6, line 130).

2. Line 187 I assume that "Me" is the abbreviation for median but it was not introduced previously in the manuscript.

The abbreviation was introduced in the sentence where it first appeared (p.8, line 187)

3. Line 210 I suppose that average should be replaced by median.

The word “average” was removed (p. 9, line 210)

4. Line 285 please write also the CI of AUC for HC vs ATB.

It was added (p. 12, line 285)

5. Line 365 and fig 5 the group ATB in the manuscript is written TB in the figure 5, could you please replace TB with ATB in the figure.

It was corrected.

6. Line 414: in the manuscript the authors mentioned “squares” as symbol for IGRA negative patients in figure 6, but I don’t see squares in figure 6, only circles of different colours. The authors should fix this point.

It was fixed (p. 18, lines 415-418)

7. Lines 488, 489, 490 and 493 please write CD4+ with the symbol + superscript.

The “+” symbol was written with the superscript (p.21, lines 490, 491, 492, 495).

To Reviewer #3: 

We would like to thank you kindly for reading our manuscript “IL-18/IL-37/IP-10 signaling complex as a potential biomarker for discriminating active and latent TB” and for all the questions and suggestions. 

Our answers to the questions are as follows:

1. - P-values in the text (lines 209 to 215, p-values are still shown with two -rather than three- decimal places)

We shown p-values in the text with three decimal places (p.9, lines 209-215).

2. - CI values in table 4 (IFN-gamma/HC vs LTBI CI)

It was added.

3. - Table 6 (decimals and n)

It was corrected.

4. Tables (e.g. tables 3 and 5) should show always the sample sizes involved in comparisons and correlations, regardless this information has already given in the text

The information on the sample sizes was added.

5. Table 7: please show the SE of the coefficients or their CI

The CI’s of the coefficients were added.

---

## [Editor Report · Decision Letter 2]

7 Nov 2019

IL-18/IL-37/IP-10 signaling complex as a potential biomarker for discriminating active and latent TB

PONE-D-19-15207R2

Dear Dr. Druszczynska,

We are pleased to inform you that your manuscript has been judged scientifically suitable for publication and will be formally accepted for publication once it complies with all outstanding technical requirements.

With kind regards,

Selvakumar Subbian, Ph.D.

Academic Editor

PLOS ONE
---

## [Editor Report · Acceptance letter]

21 Nov 2019

PONE-D-19-15207R2 

IL-18/IL-37/IP-10 signaling complex as a potential biomarker for discriminating active and latent TB 

Dear Dr. Druszczynska:

I am pleased to inform you that your manuscript has been deemed suitable for publication in PLOS ONE. Congratulations! Your manuscript is now with our production department. 

With kind regards,

on behalf of

Dr. Selvakumar Subbian 

Academic Editor

PLOS ONE